# Case Report—*Escherichia coli* Pericarditis after Recent COVID-19 Pneumonia

**Xinhang Tu \*, Nahar Saleh and Raymond Young**

Medstar Health, Baltimore, MD 21218, USA
* Correspondence: tuxinhang@hotmail.com

**Abstract:** Before the widespread use of antibiotics, purulent pericarditis was not uncommon and was predominantly a complication of bacterial pneumonia. We present a rare case of Escherichia coli (*E. coli*) purulent pericarditis without a clear source after a recent COVID-19 infection. **Background:** COVID-19 has been affecting millions of people worldwide since the outbreak in December 2019, and its involvement in multiple organ systems has been observed and studied. Multiple cardiac complications have been reported, most significantly the virus' ability to induce acute coronary syndrome and myocarditis. Pericardial disease has also been reported but less frequently. COVID-19 infection is associated with a higher risk of secondary bacterial infection, but it is rare to see purulent pericarditis in the setting of a recent COVID-19 infection. **Objective:** Our case report depicts a patient who developed *E. coli* purulent pericarditis in the setting of a recent COVID-19 infection. It indicates a possible association between COVID-19 infection with dysregulation of the immune system.

**Keywords:** COVID-19; purulent pericarditis

## 1. Introduction

A 71-year-old woman with type 2 diabetes, hypertension, a history of deep vein thrombosis (DVT) and gastrointestinal (GI) bleeding presented to the hospital with confusion and fatigue.

She was recently hospitalized with hypoxic respiratory failure secondary to COVID-19 pneumonia 2 months prior to her current presentation. She was noted to have bilateral infiltrates on non-contrast Computed tomography (CT) chest and was treated with corticosteroids, Remdesivir and cefuroxime. Her family brought her to the hospital because of worsening confusion. She had not received the COVID-19 vaccine per record.

On presentation, she looked clammy, she was afebrile (T 36.8 °C), tachycardiac (HR 107) and tachypneic (Respiry Rate 26), yet normotensive (Blood pressure 104/80), and she was saturating 100% on room air. The patient had mild hyponatremia, leukocytosis and thrombocytosis, elevated ferritin level, C-reactive protein (CRP) and Erythrocyte sedimentation rate (ESR), and elevated procalcitonin with negative high-sensitivity troponins. Her HbA1C was 7 on admission. An electrocardiogram (EKG) demonstrated sinus tachycardia with diffuse ST elevation with PR depressions with subtle electrical alternans noted (Figure 1). A bedside echocardiogram revealed a large circumferential pericardial effusion with signs of tamponade, most notably early right ventricular diastolic collapse (Figure 2). Our patient was referred for urgent pericardiocentesis with the removal of 450 cc of cloudy yellow fluid. Pericardial fluid analysis demonstrated a neutrophilic leukocytosis. She was empirically started on Piperacillin-Tazobactam and vancomycin for 2 days; culture from pericardial effusion showed pan-sensitive *E. coli*.

The patient was then admitted to the general medicine floor. A follow-up echo was obtained, which showed a normal left ventricular ejection fraction with small circumferential pericardial effusion without signs of tamponade.

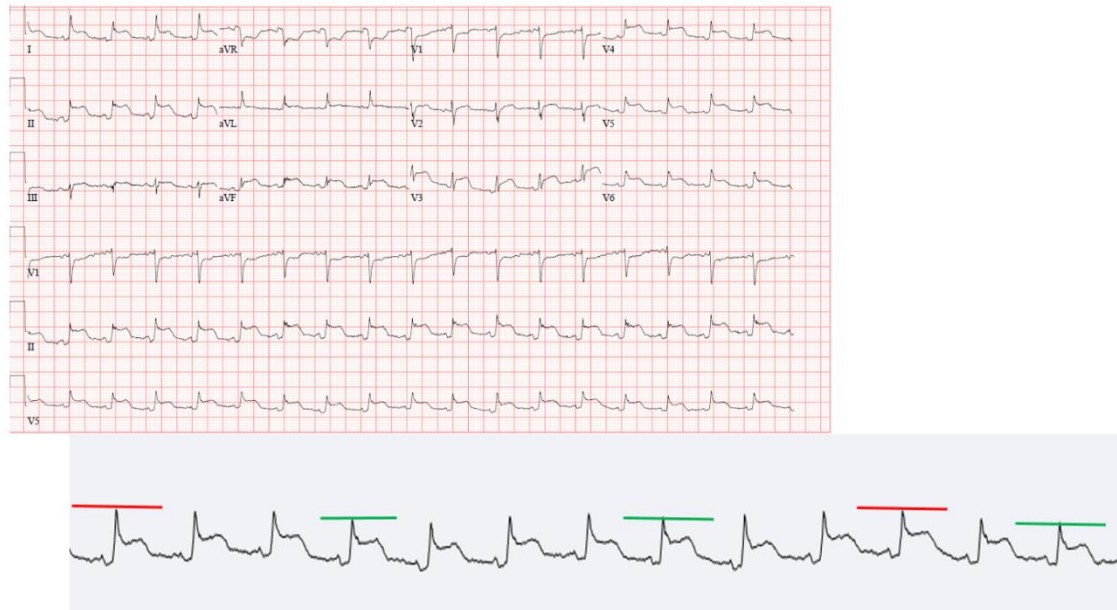

**Figure 1.** EKG on presentation.

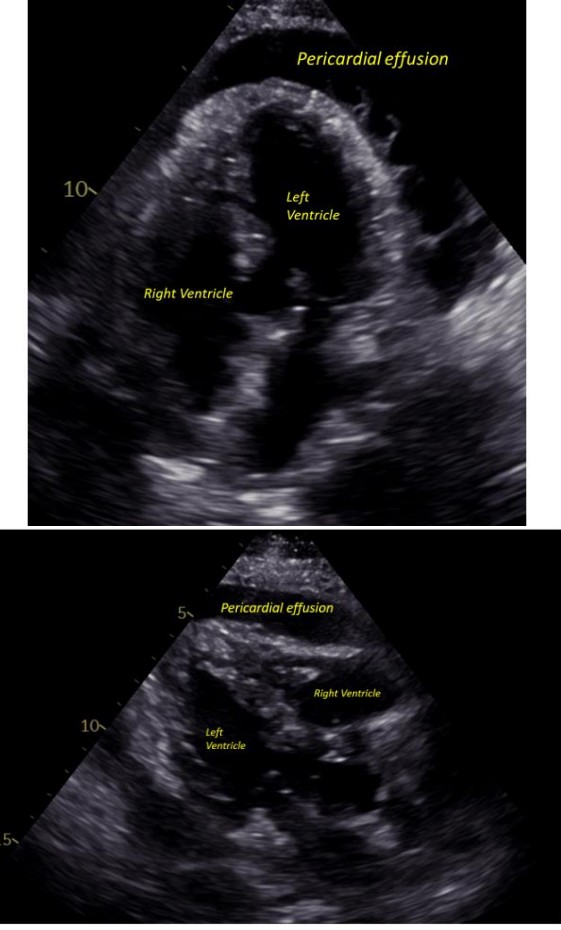

**Figure 2.** Pericardial effusion on ECHO.

Given the bacterial growth in the pericardial fluid, a search for primary infectious focus was initiated. The patient underwent a pan CT scan of the chest, abdomen and

pelvis, which failed to reveal any primary source of infection or malignancy. Blood cultures remained negative for growth, and so was a urine culture obtained on admission. A full respiratory viral panel was negative as well. Antinuclear Antibody (ANA) and anti-double stranded DNA (dsDNA) were both negative.

On day 6 of hospitalization, the patient became hypotensive, requiring an upgrade to the intensive care unit and initiating vasopressors. A bedside echocardiogram was obtained, which did not reveal any re-accumulation of pericardial fluid, but there were signs of a pericardial constriction pattern. Cardiothoracic surgery was consulted for a possible pericardial window; however, the patient was deemed a poor candidate for intervention, given her age and comorbidities. The patient was subsequently treated medically with ceftriaxone, intravenous fluids and vasopressors that were successfully weaned off within 24 h.

The patient responded well to conservative management, and she achieved full recovery. She received 8 days of ceftriaxone and was discharged with a 4-week course of doxycycline 100 mg twice daily. A follow-up outpatient colonoscopy showed diffuse melanosis coli with no polyps, strictures, masses or mucosal abnormalities.

## 2. Discussion

Purulent pericarditis is an acute, fulminant illness with a high rate of mortality and morbidity. Even with appropriate treatment, the mortality rate approaches 40% [1].

Historically, it used to be a frequent complication of pneumococcal pneumonia before the widespread use of antibiotics. In the modern era, most cases of purulent pericarditis are associated with nosocomial bloodstream infections, thoracic surgery, or an immuno-compromised state. Bacterial pericarditis accounts only for 1–2% of the etiology of acute pericarditis [2].

Pathogens causing purulent pericarditis are usually Gram-positive microbials, most commonly Staphylococcus aureus and Streptococcus pneumoniae. However, in the patients studied after 1943, Gram-negative organisms (Proteus, *E.coli*, Pseudomonas, Klebsiella etc.) counted for about one-third of the isolated organisms, and most are associated with concurrent gram-negative bacteremia, which was not the case in our patient. Patients without a clear source of the primary infection were found to have noninfectious conditions, including recent thoracic surgery, chronic renal disease, carcinoma, and diabetes [3].

### 2.1. Diabetes and Infection

Diabetes has been shown to be a risk factor for infections. A study has shown that persons with diabetes have a substantially increased risk for enterobacterial bacteremia. Among patients with bacteremia, diabetes was also associated with a poorer prognosis [4].

During the COVID-19 pandemic, evidence suggested that patients with type 2 diabetes who have COVID-19 infection are prone to have increased inflammation, based on the presence of a higher percentage of CD8+ T cells, an increase in the Th1/Th2 ratio and elevated levels of cytokines (e.g., IL-4, IL-10, IL-13, IFN-$\gamma$, and Tumor necrosis factor alpha (TNF-$\alpha$) [5]. In addition, a study in China demonstrated high HbA1c level is associated with inflammation in COVID-19 patients, and a higher mortality rate is observed in diabetic patients [6].

There appears to be a correlation between mortality in COVID-19 patients and the level of HbA1C as well—a study showed a higher COVID-19-related mortality rate in the diabetic patient population, and higher HbA1C is associated with a higher hazard ratio (2.61 when HbA1c $\geq$ 58 mmol/mol, 1.58 when <58) [7] This was supported by another population-based cohort study in diabetic patient population which showed a higher HbA1C is associated with higher COVID-19-related mortality in patients with both type 1 and type 2 diabetes [8]. However, the result of another retrospective study suggests that there was no significant association between HbA1C level and adverse clinical outcomes in diabetic patients hospitalized with COVID-19 [9]. Therefore, although it is possible that diabetes has played a role in predisposing our patient to the rare bacterial infection, the

exact effect of diabetes or HbA1C level on COVID-19 infection and its mechanism are still to be determined pending more researches.

*2.2. COVID-19 and Pericarditis*

There are several cardiac manifestations observed in COVID-19 patients, including acute myocardial injury with troponin elevation, acute heart failure, myocarditis and myopericarditis [10]. According to a meta-analysis, 4.55% of COVID-19 patients who underwent CT imaging were found to have pericardial effusions [11]. Another study showed the prevalence of pericardial effusion in a group of hospitalized patients with COVID-19 infection is around 14% based on echocardiographic examination [12]. Furthermore, severe/critical patients were found to have higher incidences of pericardial effusion [13].

There are cases reporting pericarditis being the presenting symptoms of COVID-19 [14]. It has also been reported that patients can develop myopericarditis several weeks after hospitalization due to COVID-19 infection [15].

Pericardial fluid is typically exudative in nature, suggestive of inflammation with a negative test for COVID-19. A case report from Italy demonstrated pericardial fluid analysis was positive for SARS-CoV-2 by a reverse-transcriptase-polymerase chain reaction (rRT-PCR) amplification with low viral load [16]. However, the validity of testing for pericardial fluid has not yet been confirmed [17–19].

The existence of viral respiratory tract infections has been associated with an increased risk of bacterial coinfections. As demonstrated in several clinical studies, secondary bacterial infection is a common complication in COVID-19 patients. Possible mechanisms include-viral infection causing impairment of host epithelial cells, desensitization and exhaustion of immune responses [20].

Patients with COVID-19 typically have a hyper-reactive immune response with increased levels of inflammatory cytokines and chemokines; the hyper-inflammatory and cytokine release syndrome (CRS) cause tissue damage, lung injury and acute respiratory distress syndrome (ARDS). The COVID-19 infection causes immune dysregulation with hyper-reaction and cytokine storm, and chronic stimulation of T cells potentially leads to T cell exhaustion, causing lower function and proliferative capacity and, theoretically, suppression of the immune system [21]. This could potentially predispose patients to infections such as bacterial pericarditis in our case.

In our case-given the timeline and the lack of virology proof in the drainage from pericardial effusion, it is challenging to prove the association between *E. coli* pericarditis and the previous COVID-19 infection.

However, there have been emerging cases of delayed inflammatory response that occur weeks to months after the initial infection—Multisystem Inflammatory Syndrome in Adults (MIS-A)—of which the onset time can spread over 4–12 weeks after acute COVID-19 infection. Effects on the immune system could be delayed and prolonged after the initial acute phase. Therefore, it is plausible COVID-19 infection might have triggered a series of inflammatory and immune response that lingered over weeks and predisposed our patient to this rare and complicated bacterial infection [22–25].

## 3. Conclusions

Acute pericarditis has been reported as a complication of COVID-19 infection, sometimes as an initial presenting finding. It is unclear whether COVID-19 infection will predispose patients to purulent pericarditis, but the impact of infection might cause dysregulation of the immune system and therefore increase the risk of secondary bacterial pericardial infection. More research is required to investigate the relationship between COVID-19 infection and the dysregulation of the immune system. Given the rare case in the setting of recent COVID-19 infection, we think there might be a link between COVID-19 infection and bacterial pericarditis in our patient, and the case makes an interesting contribution to medical literature.

## 4. Limitations

This case indicates a possible connection between COVID-19 infection and immune system disturbance which might have predisposed this patient to this rare *E.coli* purulent endocarditis. However, the previous COVID-19 infection was 8 weeks before her presentation, and it is extremely challenging to prove the connection between the two without supporting lab results. We hope this case report can draw attention to the effect of COVID-19 infection on the immune system in patients without baseline immunodeficiency.

## 5. Learning Objectives

a.  To be able to recognize the clinical presentation and features of pericarditis;
b.  To be familiar with the diagnostic tools and treatment modalities;
c.  To be aware that COVID-19 infection can be related to pericarditis and secondary bacterial infection, with a complicated impact on the immune system.

**Author Contributions:** Conceptualization, X.T. and R.Y.; resources, X.T. and R.Y.; writing-original draft preparation, X.T.; writing-review and editing, X.T., R.Y. and N.S.; supervision, R.Y. and N.S. All authors have read and agreed to the published version of the manuscript.

**Funding:** This research received no external funding.

**Institutional Review Board Statement:** Ethical review and approval were waived for this study and consent was obtained from the sole patient involved in this study.

**Informed Consent Statement:** Informed consent was obtained from the one subject involved in the study.

**Data Availability Statement:** The data presented in this study are as shown in the content, additional data are not publicly available due to HIPPA and authorized access required to patient's electronic medical records.

**Acknowledgments:** I would like to express my deep gratitude to Raymond Young, my supervising cardiologist, and Nahar Saleh, my prior chief resident, for their patient guidance, enthusiastic encouragement and useful critiques of this research work. I would also like to extend my thanks to my residency program, which provides full support for the case report. Finally, I wish to thank my parents for their support and encouragement throughout the study.

**Conflicts of Interest:** The authors declare no conflict of interest.

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
