# Peer review of "Case Report—Escherichia coli Pericarditis after Recent COVID-19 Pneumonia"

_2673-527X, doi:10.3390/jor3020010_

Round 1

Reviewer 1 Report

This is an interesting case reporting E-coli purulent pericarditis in a patient without clear source, presenting around 2 months after hospitalization for Covid-19 pneumonia and respiratory failure. Underlying co-morbidities of type-2 DM and history of GI bleed in this 71yr old patient are pertinent. In this interesting case, the authors discuss potential mechanisms in context of risk factors of Type 2 DM and Covid-19 infection that may have predisposed her to the bacterial pericarditis presentation. 

A few details and pointers that can strengthen this report further include:

1. Inclusion of information on ventricular function on presentation as well as on day-6 when clinical status changed requiring ICU level care. 

2. Blood counts (CBC/diff), any immune deficiency work up that may have been done.

3. Was pericardiectomy (suggested in constrictive pericarditis) being considered  or  a pericardial window (used typically in recurrent pericardial effusions) when there were signs of pericardial constriction noted on day-6?

4. Was there history of recent GI bleed/ulcer that could have been related to bacterial invasion into blood stream and subsequent bacterial pericarditis?

5. Including the HbA1c level of the patient, to put it in perspective with the discussion in the section-  diabetes and infection?

6. Did the patient have pericarditis/myopericarditis during preceding Covid-19 infection?

Author Response

Thank you for the comments, please see the reply below:

Response 1

  1. Inclusion of information on ventricular function on presentation as well as on day-6 when clinical status changed requiring ICU level care. 

-Pre-pericardiocentesis limited ECHO showed --There are signs of right ventricular early diastolic filling dysfunction, suggestive of tamponade physiology, while Left ventricular ejection fraction is 60% to 65%

  1. Blood counts (CBC/diff), any immune deficiency work up that may have been done.

-On presentation, patient had leukocytosis  (WBC 20.9 ), Hb 10.5 (baseline anemia Hb 9-10) was thought to be due to anemia of chronic disease, PLT was 580,  ANA and dsDNA were negative, HIV Ab/Ag 5 years before admission was negative, she had a colonoscopy 1 year after showed diffuse melanosis coli otherwise unremarkable.

  1. Was pericardiectomy (suggested in constrictive pericarditis) being considered  or  a pericardial window (used typically in recurrent pericardial effusions) when there were signs of pericardial constriction noted on day-6?

-Yes the case was discussed with thoracic surgery for possible surgery, yet given patient had high peri-op morbidity and high risks, the decision was made to treat conservatively after discussion.

  1. Was there history of recent GI bleed/ulcer that could have been related to bacterial invasion into blood stream and subsequent bacterial pericarditis?

-That’s a great point. Patient did have a remote history of GI bleeding without recent bleeding episodes, yet patient did not report abdominal pain/melena or other signs or symptoms concerning for GIB during this admission or prior hospitalization for covid pneumonia. Her Hb was at baseline on presentation, the CTA/P did show 1.6 cm nodule thickening and area of narrowing ascending colon is nonspecific, and the outpatient colonoscopy as mentioned above showed diffuse melanosis coli otherwise unremarkable without findings of active bleeding/malignancy.

  1. Including the HbA1c level of the patient, to put it in perspective with the discussion in the section-  diabetes and infection?

-Interestingly her HbA1C on admission was 7.

  1. Did the patient have pericarditis/myopericarditis during preceding Covid-19 infection?

-No ECHO was done during that hospitalization or prior, and per documentation there was no concern/ symptoms indicating possible pericarditis/ myopericarditis, a CT without contrast during the hospitalization for covid-19 pneumonia reported “Normal heart and pericardium. There are coronary artery calcifications.”

Reviewer 2 Report

This case report demonstrated a case of purulent pericarditis after a recent COVID infection. According to the author, this condition is rare since no identifiable primary source exists. Reviews regarding the potential mechanism were comprehensive. However, the existence of pericarditis after respiratory tract infection is not rare and not necessarily related to COVID-19 infection. The case provided limited clinical information to the readers.

Author Response

-Thank you for your comment! The rarity of this case lies in E coli pericarditis without clear source, the occurrence in a patient without immunodeficiency, and a possible connection with recent covid-19 infection. You were right-we can’t prove whether there was any cause and effect relationship between the covid-19 infection and the purulent pericarditis, yet it has been shown that covid-19 infection does lead to inflammatory response and immune system imbalance by causing a inflammatory surge, and that could potentially lead to more research on how the inflammatory reaction and immune system imbalance triggered a possible immunocompromised state of the patient infected with covid, and how long this possible immunocompromised state last even after full recovery of covid-19 infection.

Reviewer 3 Report

Pericarditis is a condition characterized by inflammation of the sac that surrounds the heart (the pericardium). Pericarditis can occur as a complication of several infectious and noninfectious causes, including viral infections such as COVID-19. There is currently no definitive percentage or an exact number of people who get pericarditis after a COVID-19 infection. However, recent studies have shown an increased risk of pericarditis and other cardiac conditions in patients who have recently contracted COVID-19. Besides, more research is needed to understand this association fully. 

The case study by Tu et al., is well-written. However, whether the patient has a history of covid-19 infection or has been vaccinated is not mentioned. 

Make the following corrections: 

In the title, write E. coli as  Escherichia coli

Line 9: Write E coli as Escherichia coli (E. coli)

Line 16: Write E coli as E. coli 

Line 61 to 65: Confusing, revise the entire paragraph. Gram-positive bacteria, such as Staphylococcus aureus and Streptococcus pneumoniae, are the most common pathogens causing purulent pericarditis. However, other microorganisms, such as fungi and gram-negative bacteria, can also cause it. So. mention the name of the bacteria. 

Author Response

Thank you for your comment and the correction of the abbreviations. Please see the attached is the revised manuscript. Thanks again.

Reviewer 4 Report

The authors present an interesting case report about purulent pericarditis two months after COVID-19 infection. While E coli pericarditis is a rare finding worth presenting, there are many major flaws in the paper that have to be cleared before publication.

First, the case report only includes limited information about the case. Vital signs as well as lab results are completely missing. Furthermore, date, type and dosages of antibiotics have to presented, together with the result of microbial resistance testing. Furthermore, the adequate dosage and duration of antibiotic therapy and alternative therapy other than pericardial drainage have to be discussed. (What were the microbiologists recommending?)

Second, I do not really see any connection with COVID-19 (by the way, which COVID strand?). In the last years we has thousands of COVID-19 infections treated in our hospitals and not a single E.coli pericarditis. I do not see any pathophysiological explanation. It is true that perimyocarditis associated with COVID-19 may occur, but this type of pericarditis is not purulent. The authors should question this connection. If the authors believe in this connection, they have to include a list of limitations in the Discussion section of the manuscript.

Minor comments:

-        Line 27: What does “stat” mean?

-        Follow Up Echo (Lines 35-37) may suite better a little bit later.

-        Section 3 “Diabetes and infection” should be a sub section of Section 2 “Discussion”.

Author Response

Thanks for your comment, please see the response below in the attached document.

Reviewer 5 Report

Xinhang Tu et al demonstrated a case report of E coli bacterial pericarditis in a 71 yo patient, 2 months after Covid 19 infection. Overall, an intersted and nice presented case report. Some minor thoughts below:

The reviewer would like to know more information about the severity of Covid 19 infection of the patient. Was the patient on mechanical or nonmechanical ventilation. How many days was she hospitalised, etc.?

I believe that the section 3. "Diabetes and infection" is a bit vague. Obviously diabetes is a bad prognostic factor regarding covid 19 mortality. First, I think there is better literature data to use as reference (not only the cohorts you mention, but also large meta analyses), and second I suggest the authors to search specifically about diabetes and pericardial effusion or pericarditis in Covid-19. Overall mortality in diabetic covid 19 patients (lines 77-86) is irrelevant to this case report.

Authors make an implication for disrupted immune response in this patient. Is there any lab value to support this hypothesis?

Please mention if the patient was vaccinated for Covid-19, when was she vaccinated and with what vaccine.

Define abbreviations upon first use properly throughout the text

Author Response

The reviewer would like to know more information about the severity of Covid 19 infection of the patient. Was the patient on mechanical or nonmechanical ventilation. How many days was she hospitalised, etc.?

-Patient was hospitalized for 7 days-- initially was put on nonrebreather in ED later transitioned to high flow nasal cannula, received 10-day dexamethasone and 5-day course of remdesivir, also was treated with cefuroxime for possible associated bacterial pneumonia yet the blood cultures were negative. She was gradually weaned to room air and on as needed 2 L nasal cannula on discharge to facility.

I believe that the section 3. "Diabetes and infection" is a bit vague. Obviously diabetes is a bad prognostic factor regarding covid 19 mortality. First, I think there is better literature data to use as reference (not only the cohorts you mention, but also large meta analyses), and second I suggest the authors to search specifically about diabetes and pericardial effusion or pericarditis in Covid-19. Overall mortality in diabetic covid 19 patients (lines 77-86) is irrelevant to this case report.

-Thanks for your comment. There is one retrospective study showed that the difference between the incidences of pericardial effusion in covid-19 patient with diabetes and without diabetes does not have statistic significance(Table1). https://www.ahajournals.org/doi/epub/10.1161/JAHA.121.024363

Authors make an implication for disrupted immune response in this patient. Is there any lab value to support this hypothesis?

-Patient had leukocytosis(20.9) and thrombocytosis (580), elevated ferritin level(726.7) , CRP (282)and ESR(>120), and elevated procalcitonin (4.54) that indicated ongoing systemic inflammation and active infection, yet they are not specific.

Please mention if the patient was vaccinated for Covid-19, when was she vaccinated and with what vaccine.

-The record indicated she did not receive covid-19 vaccine before her recent covid-19 infection.

Round 2

Reviewer 2 Report

The author has addressed the rarity of this case again in the reply. The case provides clinical awareness of pericarditis as a possible diagnosis after COVID infection. 

Author Response

Thanks for your comment!

Reviewer 4 Report

Unfortunately, the authors did not adapt the manuscript accordingly to my response. For such a case report, the authors have to include the exact lab values and the results from the microbial resistance testing. Furthermore, the exact course of antibiotics is essential to be included in the main manuscript. To my knowledge, there is no character limit for case report. Therefore, please include this information into the manuscript.

Author Response

Thanks for the comments. Please see the revised draft.